

# Bacterial microbiome profiles of the inflamed terminal ileum mucosa in active Crohn's disease patients

Juan Yin[1], Tong Hu[2], Liping Zhang[2], Lijuan Xu[2], Jianyun Zhu[1], Yulan Ye[2] and Zhi Pang[1,2]

[1] Department of Digestive Disease and Nutrition Research Center, The Affiliated Suzhou Hospital of Nanjing Medical University, Suzhou Municipal Hospital, Gusu School, Nanjing Medical University, Suzhou, Jiangsu, China

[2] Department of Gastroenterology, The Affiliated Suzhou Hospital of Nanjing Medical University, Suzhou Municipal Hospital, Gusu School, Nanjing Medical University, Suzhou, Jiangsu, China

Corresponding authors
Yulan Ye, yeyulan59@163.com
Zhi Pang, pangzhi0273@sina.com

## ABSTRACT

**Background**. The dysbiosis of the intestinal microbiome relates to the pathogenesis of Crohn's disease (CD). Previous studies have focused on microbiota diversity and composition in CD patients in comparison with healthy individuals. Nonetheless, intestinal flora varies greatly among individuals. This study aimed to characterize the bacterial microbiome profiles of inflamed mucosa in comparison with those of noninflamed mucosa from Crohn's disease patients.

**Methods**. Sequencing of 16S rDNA V4 was used to identify the bacterial microbiome profiles of twelve pairs of inflamed and noninflamed mucosa from active CD patients.

**Results**. A total of 753 OTUs were specific to inflamed tissues. Alpha diversity demonstrated that the biodiversity of the bacterial microbiota in the inflamed mucosa was increased, but it was not significantly different from that in the noninflamed mucosa. Linear discriminant analysis effect size (LEfSe) Clustergram showed *Micrococcaceae*, *Bifidobacteriaceae*, *Bifidobacteriales*, *Flavobacteriaceae*, and *Methylobacteriaceae* as microbes that were significantly different in the inflamed mucosa of active CD patients. Wilcox test results of genus differences indicated *Methylobacterium*, *Rothia*, *Shinella*, *Capnocytophaga*, *Actinomyces*, *Gardnerella*, *Leucobacter*, and *Bifidobacterium* as significantly upregulated genera in the inflamed mucosa of active CD patients compared with their noninflamed mucosa. These findings provide new evidence that the dysbiosis of mucosa-associated microbiota contributes to CD development, from a self-comparison perspective.

## INTRODUCTION

Inflammatory bowel diseases (IBDs) which include Crohn's disease (CD) and ulcerative colitis (UC), have become global diseases with accelerating incidence. Patients with IBD showed an abnormal mucosal immune response and compromised epithelial barrier function (*Ng et al., 2017*). Unlike ulcerative colitis, Crohn's disease demonstrates transmural inflammation with skip lesions, primarily affecting the terminal ileum

and proximal colon in a segmental manner (*Dolinger, Torres & Vermeire, 2024*). The pathogenesis of CD is complex and inexplicable. Most existing research suggests that Crohn's disease might result from dysregulated adaptive and innate immune responses owing to a complex interplay between altered gut microbiota, environmental factors, and genetic susceptibility (*Torres et al., 2017*). With increasing urbanization, the incidence of CD has climbed in the Asia-Pacific region over the last decade (*Ng et al., 2019*). The excessive pursuit of cleanliness in industrial processes may lead to changes in the interaction between environmental microorganisms and the intestinal tract. Therefore, it is significant and urgent to study the gut bacterial microbiome to discover the pathogenesis of CD.

Recent studies on the gut microbiome of IBD patients have made progress in understanding both the composition and function of microbes in IBD progression. An increasing number of studies have focused on differences in intestinal flora composition in mucosa or feces of IBD patients in comparison with healthy controls, in disease progression (*Assa et al., 2016*) or treatment (*He et al., 2019*). In general, reduced diversity of microbiota with the shift in the abundance of particular taxa was present in IBD patients (*Kudelka et al., 2020*). Reduced abundance of *Lactobacillus*, *Firmicutes*, *Bacteroides*, *Clostridia*, *Bifidobacterium*, *Faecalibacterium prausnitzii*, and *Ruminococcaceae*, and increased levels of *Fusobacterium* species, *Escherichia coli*, and *Gammaproteobacteria*, were discovered in IBD patients (*Kostic, Xavier & Gevers, 2014*; *Sokol et al., 2017*). Regarding treatment-naïve CD patients, a large cohort study demonstrated an increased abundance of *Enterobacteriaceae*, *Veillonellaceae*, *Fusobacteriaceae*, and *Pasteurellaceae*, and a decreased abundance of *Erysipelotrichales*, *Clostridiales*, and *Bacteroidales*, which was correlated with CD disease activity (*Gevers et al., 2014*). Fecal microbiota transplantation could lead to an expansion of intestinal microbial diversity following treatment of active Crohn's disease (*Vaughn et al., 2016*).

However, the intestinal flora varies greatly among individuals. Recent studies have turned to individual gut flora variation (*Braun et al., 2019*). Some studies focused on the microbial community between the inflamed and adjacent uninflamed tissue sites from CD patients. Several studies demonstrated that the microbiome was similar in the inflamed and noninflamed mucosa (*Gao et al., 2023*; *Olaisen et al., 2021*). Whereas, there was a study that suggested that noninflamed tissues formed an intermediate population between controls and inflamed tissues. Species richness increased from control to noninflamed tissue, and then declined in fully inflamed tissue (*Sepehri et al., 2007*). The number of studies investigating the pathogenic mechanism of CD in connection with intestinal flora has increased. Thus, this study aimed to characterize the bacterial microbiome profiles of inflamed mucosa in comparison with noninflamed mucosa in Crohn's disease patients. This may provide new evidence in support of the point that the dysbiosis of mucosa-associated microbiota contributes to CD development, from a self-comparison perspective.

## MATERIALS & METHODS

Portions of this text were previously published as part of a preprint (https://doi.org/10.21203/rs.3.rs-2449245/v1).

**Table 1  Characteristics of enrolled patients.**

| Patient no | Sex (M/F) | Age (y) | WBC ($10^9$/L) | CDAI (scores) | CALP (μg/g) | ESR (mm/h) |
|---|---|---|---|---|---|---|
| 1 | M | 30 | 3.77 | 281.2 | >1,000 | 23 |
| 2 | M | 27 | 3.4 | 160.2 | 51.4 | 6 |
| 3 | F | 52 | 5.9 | 151.4 | 25.6 | 16 |
| 4 | F | 35 | 4.95 | 237.9 | 240 | 31 |
| 5 | M | 42 | 5.69 | 159 | 42.2 | 9 |
| 6 | F | 40 | 5.9 | 162.3 | 15 | 16 |
| 7 | F | 42 | 4.91 | 251.6 | 50.4 | 31 |
| 8 | M | 29 | 5.82 | 150.2 | 9 | 1 |
| 9 | F | 53 | 4.11 | 161.9 | 20.1 | 18 |
| 10 | M | 50 | 5.24 | 163.9 | 6.8 | 4 |
| 11 | F | 53 | 4.23 | 154 | 38.2 | 20 |
| 12 | F | 33 | 7.63 | 293.7 | 287 | 26 |

**Notes.**
WBC, white blood cell count; CALP, fecal calprotectin.
Crohn's disease activity index;
y, year; M, male; F, female.

## Recruitment of participants

Twelve CD patients with ileal and colon inflammation in the active phase (CD activity index, CDAI > 150) who were not undergoing treatment were enrolled in this study. Samples of patients with CD were collected from January to November 2019 at the Affiliated Suzhou Hospital of Nanjing Medical University, Suzhou Municipal Hospital. The characteristics of enrolled patients are shown in Table 1. Patients who accepted therapeutic antibodies and antibiotic drug treatment for six months were excluded. Ethical approval for this study was obtained from the Ethics Committee of Nanjing Medical University. All experiments were performed in accordance with relevant guidelines and regulations by the Research Ethics Boards at the Ethics Committee of Suzhou Municipal Hospital, Nanjing Medical University (NO. K2021-064-K01). Written informed consent was obtained from all participants.

## Collection of terminal ileum mucosal samples

The samples were collected as part of the routine diagnostic workflow. Only a subset of these samples was utilized for research purposes. Two inflamed mucosal biopsies were collected from the terminal ileum of CD patients. Two noninflamed mucosal biopsies located 10 cm from the inflamed mucosa of the terminal ileum were collected. Biopsy specimens were immediately immersed either in liquid nitrogen for rapid freezing (one inflamed and one noninflamed mucosal biopsy from each patient with CD) or in 4% formaldehyde (one inflamed and one noninflamed mucosal biopsy from each patient with CD) as they underwent the colonoscopy. The specimens that were frozen in liquid nitrogen were stored at −80 °C before DNA extraction and library construction. The biopsies fixed in 4% formaldehyde were subjected to HE staining. The degree of tissue inflammation was identified by HE staining.
### DNA extraction, library construction, and 16S rRNA gene sequencing

A total of 24 mucosal biopsies were collected from CD patients. Genomic DNA was extracted with a QIAamp$^R$ DNA mini kit (No. 51306; Qiagen, Hilden, Germany) according to the manufacturer's instructions.

A total of 30 ng of qualified genomic DNA samples and their corresponding fusion primers were selected for PCR, and parameters were set for amplification. Furthermore, Agencourt AMPure XP magnetic beads were used to purify the PCR amplification products and subsequently dissolve them in an elution buffer, which was labeled to complete the database construction. An Agilent 2100 BioAnalyzer was used to detect the range and concentration of fragments in the library. The HiSeq platform was selected for sequencing of qualified libraries according to the size of the inserted fragments.

### Data analysis of microbial load
#### Tag connections

The offline data were filtered. The remaining high-quality clean data were used for further analysis. Reads were stitched by overlap relationships between read tags using FLASH (Fast Length Adjustment of Short reads v1.2.11).

### OTU clustering and analysis

Tags were clustered into operational taxonomic units (OTUs) by USEARCH (v7 .0.1090) to compare with the gold database (v20110519) for chimeric removal. The Usearch global method was used to calculate the abundance of OTUs. The following analyses were conducted to characterize OTUs, including an OTU Venn diagram, principal components analysis (PCA), species accumulation curves, partial least squares discrimination analysis, PLS-DA, and OTU rank curve.

### Species annotation, composition, and biodiversity analysis

Species annotation was carried out by comparing the OTU representative sequence with the database through RDP Classifier (V2.2) software. The reliability threshold was set to 0.8To obtain the species classification information corresponding to each OTU, the RDP classifier Bayesian algorithm was used to classify the OTU representative sequences. The community composition of each sample was counted at all levels including phylum, class, order, family, genus, and species.

The species diversity of a single sample was analyzed by alpha diversity. Alpha diversity analysis was performed using MOTHUR (version 1.31.2). The observed species index, chao index, ace index, Shannon index, simpson index, and good-coverage index are shown to describe diversity. The first four indices previously listed are directly correlated with species abundance, such that a greater value indicates a greater abundance of species in the sample. However, the fifth index listed (good-coverage index) has the opposite relationship, so a smaller value indicates more abundant species in the sample. In other words, as the value of the good-coverage index increases, the probability that the sequence in the sample will not be measured decreases. The difference in alpha diversity between groups is displayed in the alpha diversity box chart. The Wilcoxon test was used for comparisons between the two groups.
Beta diversity analysis was used to compare the degree of difference in species diversity between pairs of samples. Beta diversity was performed through QIIME (v1.80). The heatmap of beta diversity is shown to reflect the similarity of bacterial species between the samples. A box chart was generated by the ggplot package of R (v3.4.1) to display differences in beta diversity between groups.

## Analysis of microbiome phylogeny

A phylogenetic branch tree was built at five levels including phyla, class, order, family, and genus (the top 50 most abundant were selected) using FastTree (version 2.1.3). The phylogenetic tree is ordered from the inside to the outside, and each circle represents a level, followed by the phylum, class, order, family, and genus levels. Individual phyla are differentiated by color, and node size is representative of species abundance. The outer ring is the abundance heatmap, each ring is a set of samples (a sample), each set of samples (each sample) corresponds to a color, and the color depth varies with species abundance. The length of the branch indicates the difference in evolutionary distance. The closer the evolutionary relationship is, the closer the evolutionary tree species is.

## Differential bacterial composition analysis between groups

Linear discriminant analysis effect size (LEfSe) software was used to discover microbes that were significantly different (https://huttenhower.sph.harvard.edu/galaxy/) (*Segata et al., 2011*). LEfSe was applied to construct a cluster diagram to determine the differential bacterial composition between groups. Taxonomic analysis at the order level demonstrated that all significant associations represented entire orders, with no significant signals coming from unclassified members within these taxa. An LDA diagram is shown to display microbes with significant differences. The Wilcoxon rank-sum test was conducted to calculate the significant difference between groups.

## Enterotype analysis

The Jensen–Shannon distance and PAM (partitioning around medoids) were calculated according to the relative abundance at the genus level for clustering. The optimal clustering $K$ value was calculated by the Calinski–Harabasz (CH) index. Then, between-class analysis (BCA, $K \geq 3$) was employed for visualization. The software R (v3.4.1) cluster and clusterSim packages were used in this analysis. Different samples with similar dominant flora structures can be grouped in this analysis, yet this is only suitable under certain conditions; these conditions include bacterial group typing of specific environmental samples, such as enterotypes, vaginal typing (cervicotype), and oral typing.

## Function prediction and correlation analysis

Kyoto Encyclopedia of Genes and Genomes (KEGG) functional prediction of bacteria with upregulated abundance in inflamed mucosa was carried out by PICRUSt2. The Wilcoxon test was used to discover functional differences between the groups.

The different bacteria at the genus level were obtained by the rank sum test, and a Spearman correlation heatmap among dominant species was generated by R software. The map reveals important patterns and relationships among dominant species.

## RESULTS

### Characteristics of participants

The clinical characteristics of the participants in this study are shown in Table 1. Twelve CD patients ranging from 27 to 53 years old, including seven women (mean age 35.6) and five men (mean age 44) were enrolled.

### OTU clustering and analysis

A total of 1,158 OTUs were shared between inflamed tissues and noninflammatory mucosa tissues (self-controls) of CD patients. Compared to self-control, 753 OTUs were specific to inflamed tissues (Fig. 1A). OTU Core-Pan displays common and unique OTUs for all samples in petal plots. Three OTUs were common in all tissues (Fig. 1B). Species accumulation curves showed an upward trend at the end of the curve tended to be flat, which indicated that the sampling amount was sufficient (Fig. 1C). Partial least squares discrimination analysis (PLS-DA) (Fig. 1D) showed the results could be divided into two distinct groups between inflamed tissues and noninflamed tissues of CD patients based on differences in the abundance of OTUs. The abscissa was sorted by sample OTU abundance (from high to low), and the ordinate is OTU abundance. The abundance of species in the sample is reflected by the length of the horizontal axis of the curve. As the curve widens, the abundance in species composition of the sample increases. The uniformity of the species in the sample is reflected by the shape of the vertical axis of the curve. If the curve is flatter, the uniformity of the species composition in the sample is greater (Fig. 1E).

### Bacterial microbiota composition and correlations in inflamed mucosa

Histograms at the genus level showed that the main genus in both inflamed tissues and noninflamed tissues (control) of CD patients, included *Escherichia* (0.2771, 0.2904) (relative abundance of the genus in inflamed tissues and noninflamed tissues), *Bacteroides* (0.1212, 0.1571), *Prevotella* (0.0734, 0.0688), *Fusobacterium* (0.0663, 0.0847), *Acinetobacter* (0.0390, 0.0323), *Pseudomonas* (0.0294, 0.0315), *Ruminococcus* (0.0204, 0.0283), *Megamonas* (0.0184, 0.0241), *Sphingomonas* (0.0176, 0.0217) and *Dorea* (0.0162, 0.0350), which collectively composed the predominant bacterial microflora (Fig. 2A). Those species whose abundances were less than 0.5% of all samples were classified into others. Enterotype analysis showed that *Enterobacteriaceae* (*Escherichia*), *Moraxellaceae* (*Acinetobacter*), and *Pseudomonadaceae* (*Pseudomonas*) were the three principal components in the mucosa of CD patients (Fig. 2B). The abscissa represents principal component one, and the ordinate represents principal component two, which are the two principal components with the largest variance contribution rate.

According to the species composition graph, the intestinal mucosa of CD patients was rich in *Bacteroidaceae* (*Bacteroides*), *Prevotellaceae* (*Prevotella*), *Fusobacteriaceae* (*Fusobacterium*), and *Enterobacteriaceae* (*Escherichia*) (Fig. 2C). A phylogeny tree (Fig. 2D) shows that *Actinobacteria*, *Bacteroidetes*, *Firmicutes*, *Proteobacteria, Verrucomicrobia, Nitrospirae*, and *Acidobacteria* were the dominant phyla in the intestinal mucosa of CD patients. The biodiversity of bacterial microbiota in inflamed mucosa was not significantly different compared with noninflamed mucosa.

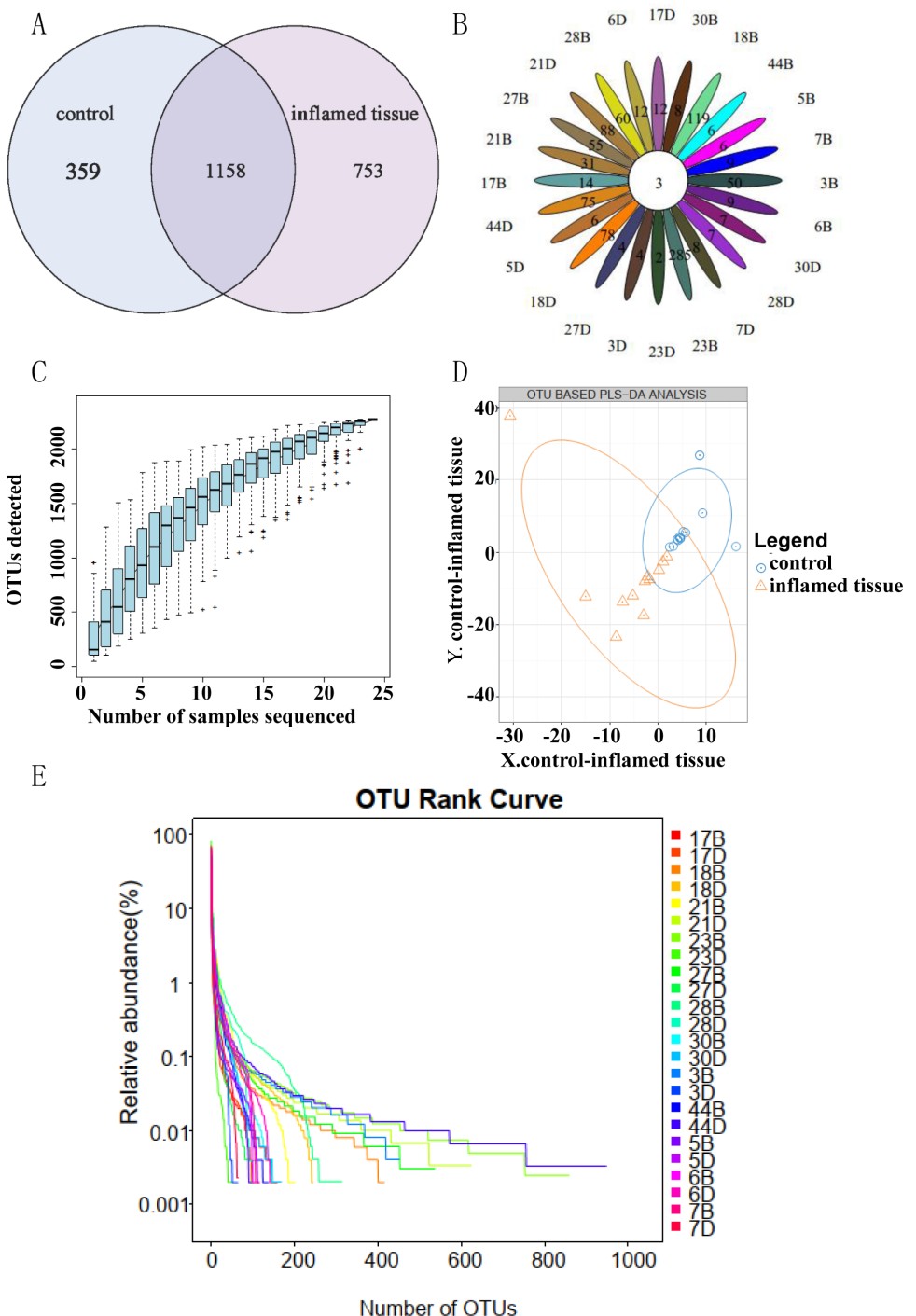

**Figure 1  OTU clustering and analysis.** (A) OTU Venn diagram showing shared and unique OTUs between groups. (B) OTU Core Pan displayed common and unique OTUs for all samples in petal plots. (C) Taxonomic accumulation curves. (D) PLS-DA by abundance differences of OTUs. The blue circles and orange triangles represent noninflamed control tissues and inflamed tissues, respectively. (E) OTU_Rank. A flatter curve represents a higher degree of uniformity in taxonomic composition from the sample.

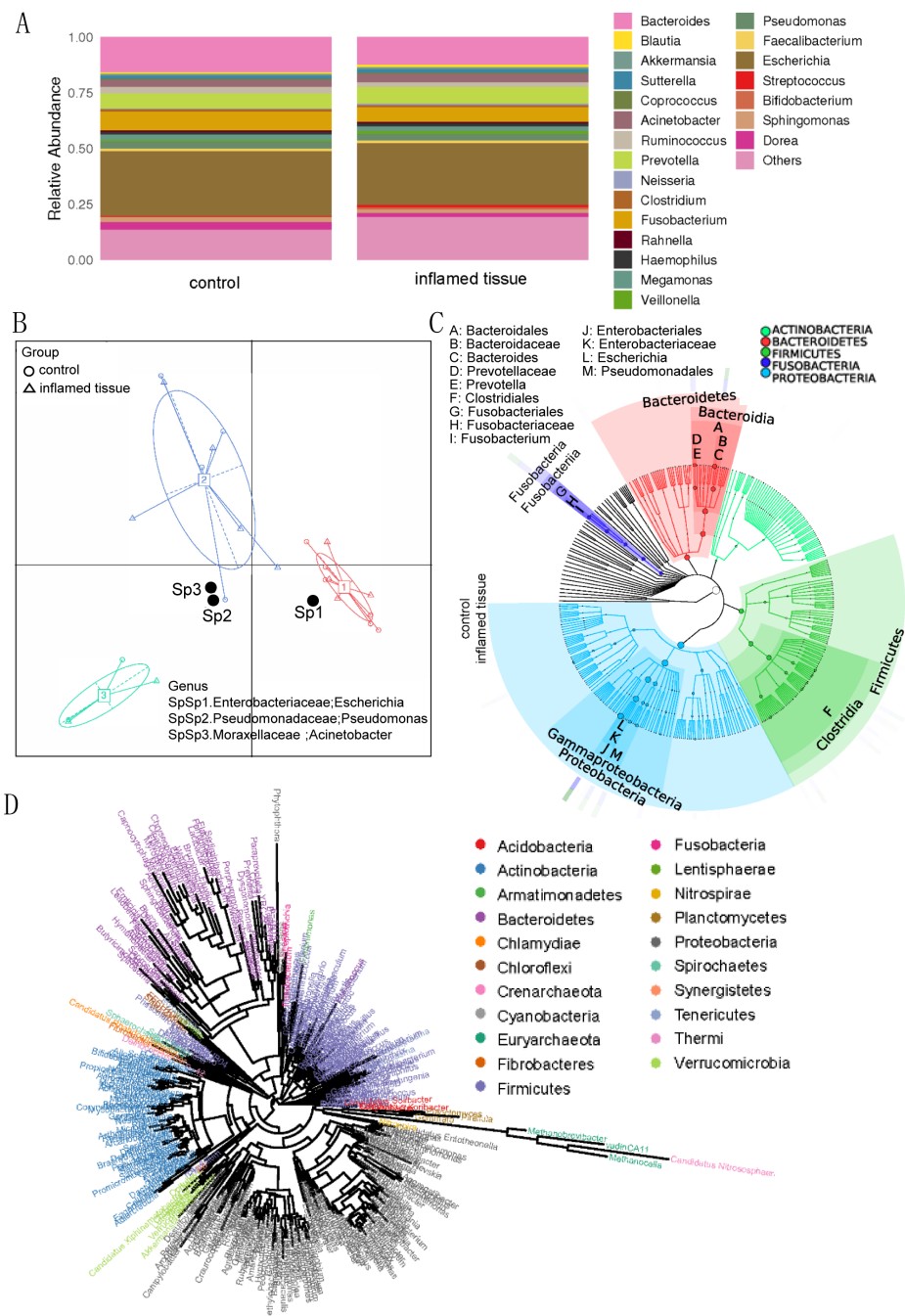

**Figure 2** **Bacterial microbiota composition and correlations in the inflamed mucosa.** (A) Bar plot. Histograms of relative abundance at the genus level. (B) Enterotype analysis showed that *Enterobacteriaceae* (*Escherichia*), *Moraxellaceae* (*Acinetobacter*), and *Pseudomonadaceae* (*Pseudomonas*) were the three principal components in the mucosa of CD patients. (C) Graphlan. According to the species composition graph, the intestinal mucosae of CD patients were rich in *Bacteroidaceae* (*Bacteroides*), *Prevotellaceae* (*Prevotella*), *Fusobacteriaceae* (*Fusobacterium*), and *Enterobacteriaceae* (*Escherichia*). (D) Phylogenetic tree of genera. The length of the branch indicates the difference in evolutionary distance. As the distance within the evolutionary tree shortens, the degree of evolutionary relationship strengthens.

Alpha diversity analysis demonstrated that the biodiversity of the bacterial microbiota in the inflamed mucosa was increased, but it was not significantly different from that in the noninflamed mucosa (Fig. 3A). The mean, SD, and *P*-value of the observed species, as well as the Chao, ACE, Shannon's diversity, Simpson's diversity, and Good's coverage, are shown in Table 2.

The beta diversity index box plot (Fig. 3B) and beta diversity matrix heatmap (Fig. 3C) showed that there were no significant differences between the noninflamed mucosa and inflamed mucosa of patients with CD.

## Differential bacterial composition in inflamed mucosa

Heatmap of bacterial relative abundance in inflamed and noninflamed mucosa (control) at the genus level, which indicated the similarity and variation in the community composition of the samples (Fig. 4A). Genera clustered on the upper-left quadrant exhibited declining abundances from control to inflamed tissue. And genera clustered in the lower-left quadrant exhibited progressively increasing abundances from control to inflamed tissue. The LEfSe Clustergram showed *Micrococcaceae*, *Bifidobacteriaceae*, *Bifidobacteriales*, *Flavobacteriaceae,* and *Methylobacteriaceae* as potential microbiota significantly contributing to group differences between the noninflamed and inflamed mucosa of CD patients (Fig. 4B). A colored dot represents a significantly different microbe, and the name of the corresponding microbe is shown in the legend on the top right. The yellow nodes represented groups of microbes that did not play an important role in the different mucosa. Working from inside toward the outside, each circle represented the species at the phylum, class, order, family, and genus levels. The LEfSe-LDA showed *Bifidobacteriales*, *Bifidobacterium*, *Bifidobacteriaceae*, *Micrococcaceae*, *Flavobacteriaceae*, *Gardnerella*, *Methylobacteriaceae*, *Rothia*, *Methyobacterium*, *Leucobacter*, *Actinomyces*, *Shinella*, and *Capnocytophaga* as potential microbiota significantly contributing to group differences between the noninflamed and inflamed mucosa of CD patients. The default preset value was 2.0 (only the absolute value of LDA greater than two will be shown in the figure). The colors in the bar chart distinguished each group, while the length represented the LDA score. This score indicated the impact size of the species with significant differences between groups (Fig. 4C). The Wilcox test results that were used to assess differences in genera indicated that the relative abundance of *Methylobacterium*, *Rothia*, *Shinella*, *Capnocytophaga*, *Actinomyces*, *Gardnerella*, *Leucobacter*, and *Bifidobacterium* was significantly increased in the inflamed mucosa of CD patients compared with the noninflamed mucosa. The histogram of the relative abundance of each group was shown on the left. Listed in the center is the log2 value of the average relative abundance ratio of the same genus between the two mucosa. The figure on the right showed the *P*-value and FDR values obtained by the Wilcox test. If the *P*-value and FDR values were less than 0.05, the genus differed significantly between the two groups (Fig. 4D).

## Functional prediction of bacteria with upregulated abundance in inflamed mucosa

Function prediction *via* KEGG indicated that bacteria in the mucosa of CD patients participated in functions such as amino acid transport and metabolism, translation,

Yin et al. (2025), *PeerJ*, DOI 10.7717/peerj.19959

**Table 2  Alpha test result.**

|  | Sobs | Chao | Ace | Shannon | Simpson | Coverage |
|---|---|---|---|---|---|---|
| CD control | 228.16667 ± 274.56373 | 247.4005 ± 302.67222 | 254.251 ± 306.33239 | 2.34758 ± 0.98971 | 0.28408 ± 0.20126 | 0.99899 ± 0.00195 |
| CD inflamed tissue | 295.91667 ± 232.15961 | 318.29943 ± 243.24375 | 336.38184 ± 248.33517 | 2.80409 ± 0.92624 | 0.21524 ± 0.16151 | 0.99919 ± 0.00094 |
| *p* value | 0.12769 | 0.10053 | 0.05966 | 0.31859 | 0.34736 | 0.26013 |

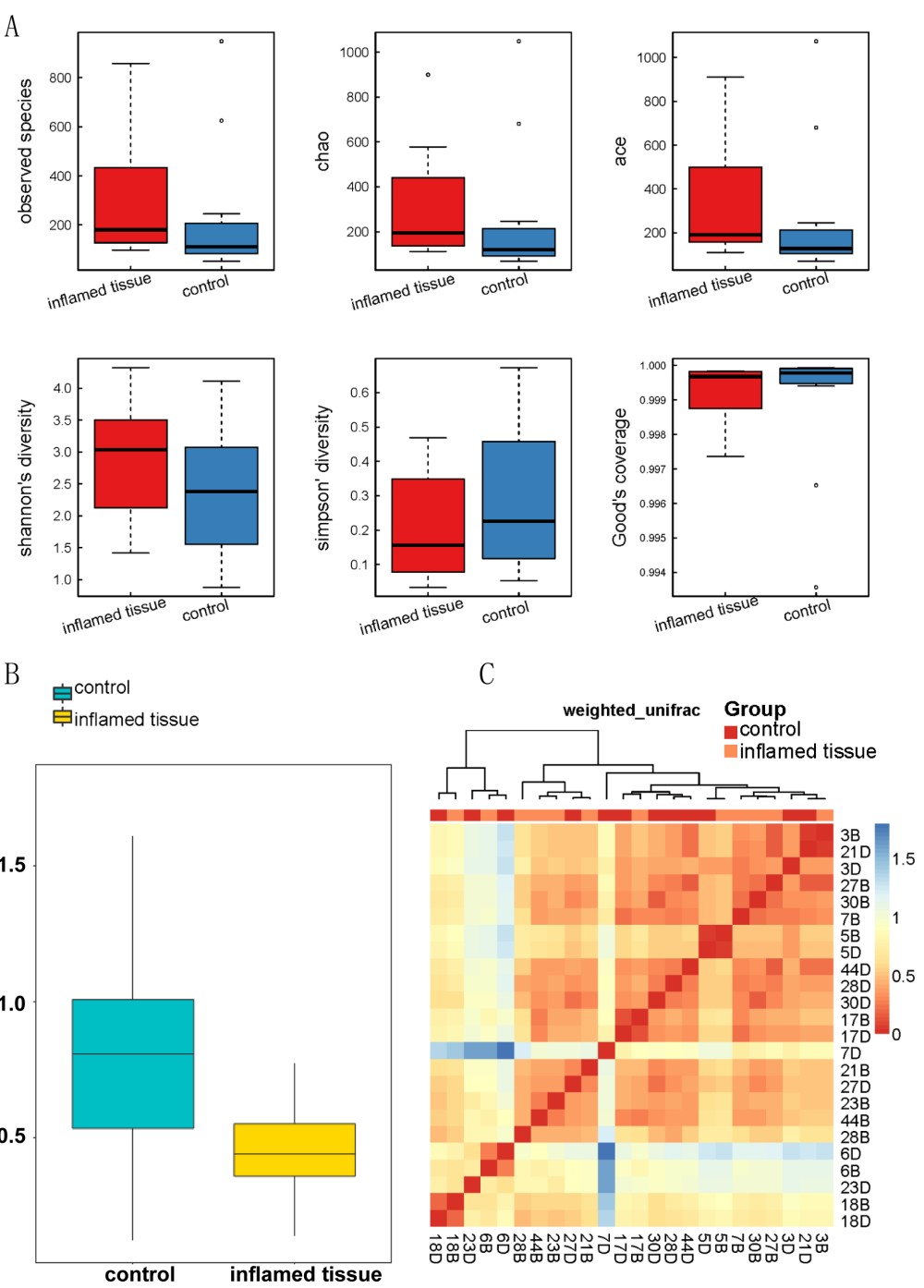

**Figure 3  Biodiversity of bacterial microbiota in inflamed mucosa.** (A) Alpha_Box. Alpha diversity demonstrated that the biodiversity of the bacterial microbiota in the inflamed mucosa was increased, but it was not significantly different from that in the noninflamed mucosa. Alpha diversity analysis was performed using MOTHUR (version 1.31.2). The mean, SD, and *p*-value of the observed species, along with the Chao, ACE, Shannon's diversity, Simpson's diversity, and Good's coverage are shown in Table 2. (B) Beta diversity index box plot. The statistical results of the differences in beta diversity between groups 

**Figure 3 (...continued)**
were used to draw a box graph to better illustrate the differences in beta diversity between groups. The R (v3.4.1) ggplot package was used to analyze the beta diversity. (C) Beta diversity matrix heatmap. The following figure is the beta diversity matrix heatmap, which visualizes the beta diversity data through graphics, and the samples with similar beta diversity are clustered together. The NMF package of R (v3.2.1) was used to analyze the beta diversity matrix heatmap.

ribosomal structure, and biogenesis, general function prediction only, carbohydrate transport and metabolism, cell wall/membrane/envelope biogenesis, transcription, energy production and conversion (Fig. 5A). KEGG functional prediction shows the predicted function of the colony based on the KEGG database. The horizontal axis represents the sample and the vertical axis represents the relative abundance of the predicted function.

Functional prediction of bacteria with upregulated abundance in inflamed mucosa indicated that there were no significantly different pathways between these two groups. (Fig. 5B). The left side shows the relative abundance histogram of each group; the middle is the log2 value of the relative abundance mean ratio of the same channel between the two groups; the $p$-value and FDR value obtained by the Wilcox test are listed on the right. If the $P$-value and FDR value were less than 0.05, then the pathway was considered significantly different between the two groups.

## Spearman correlation coefficient analysis of species

The genera were differentiated by the rank sum test, and a heatmap showing the Spearman correlation among dominant genera was generated by R software (Fig. 6A). Through this heatmap, important patterns and relationships among dominant genera can be found. The graph shows correlations between genera. A darker color indicates more closely correlated. The strongly positively correlated (correlation coefficient ≥0.6) bacterial genera were *Dorea-Ruminococcus* (0.65) (correlation coefficient), *Haemophilus- Fusobacterium* (0.62), *Megamonas-Faecalibacterium* (0.64), *Blautia-Dorea* (0.76), *Blautia-Sutterella* (0.64), *Akkermansia-Ruminococcus* (0.60), *Coprococcus-Dorea* (0.6), *Coprococcus-Blautia* (0.8), *Neisseria-Fusobacterium* (0.64), and *Neisseria-Haemophilus* (0.64). There were no strongly negatively correlated (absolute value of correlation coefficient ≥ 0.6) genera. The top 5 of the genera with moderate negative correlation (absolute correlation coefficient ≥0.3 and <0.6) were *Akkermansia-Veillonella* (−0.55), *Coprococcus-Escherichia* (−0.47), *Rahnella,-Haemophilus* (−0.46), *Clostridium-Megamonas* (−0.46), and *Sphingomonas- Pseudomonas* (−0.45). Genera that were negatively correlated include. Figure 6B showed the network map of genera. Each node in the graph represented an individual genus. The size of the node represented the magnitude of the average relative abundance of the genera. Straight lines were used to link genera, with pink indicating a positive correlation, blue indicating a negative correlation, and the thickness of the lines indicating the magnitude of the correlation. This figure only showed the relationship between genera with a correlation coefficient greater than 0.2.

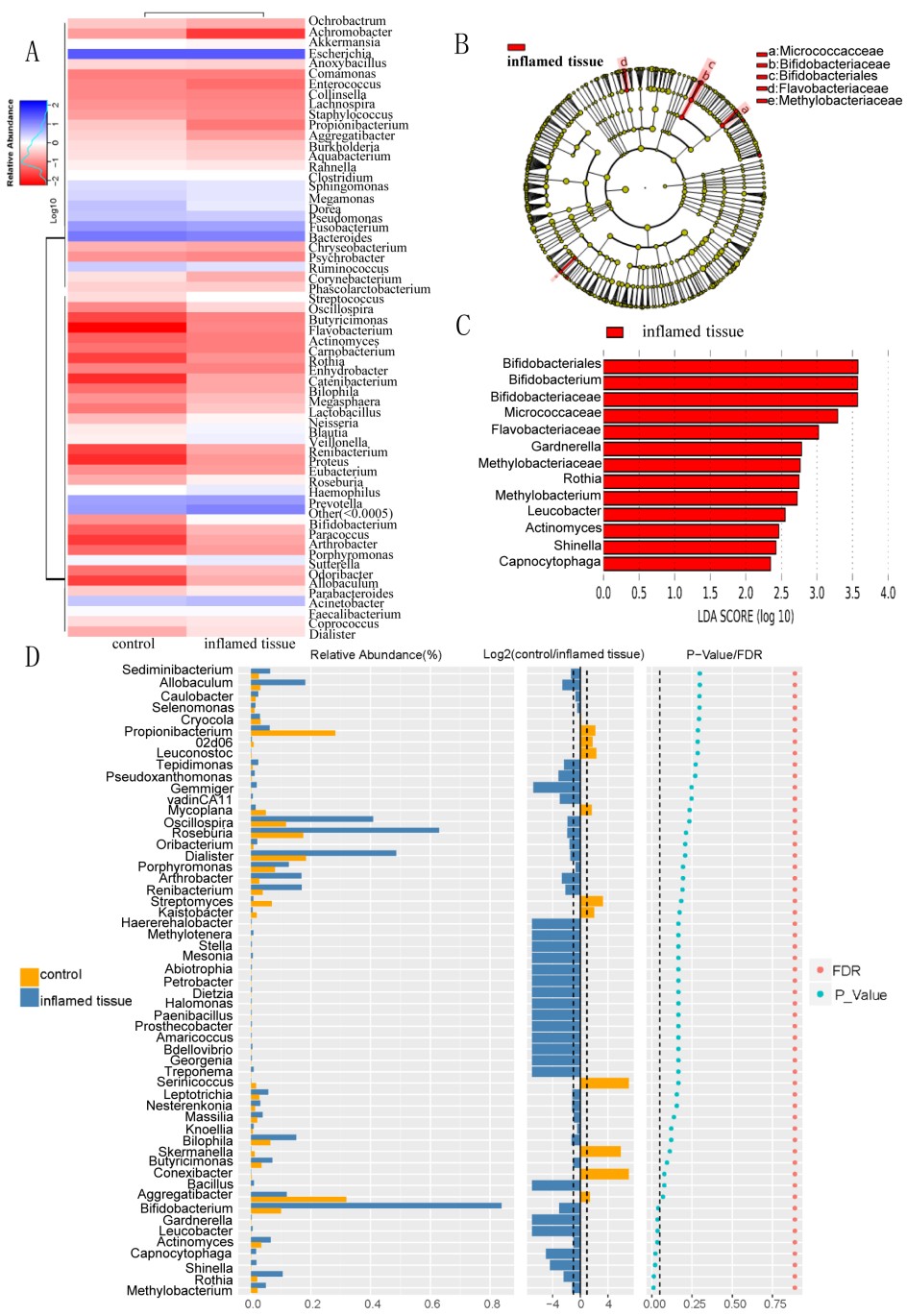

**Figure 4 Differential bacterial composition in the inflamed mucosa.** (A) Heatmap of bacterial relative abundance in inflamed and noninflamed mucosa at the genus level. Geunus with abundances <0.5% across all samples were grouped as "Others". (B) LEfSe and (C) LEfSe-LDA showed bacterial with significant differences in the inflamed mucosa of CD patients. Potential microbiota significantly contributing to group differences were mainly present in orders with significant differences in LDA scores greater than the value. The default preset value was 2.0 (only the absolute values of LDA greater than 2 are shown in this figure). The color of the bar chart represents each group, while the length represents the LDA score, (continued on next page...)

**Figure 4 (…continued)**
which indicates the impact size of the orders with significant differences between different groups. (D) The Wilcoxon test results indicated different genera were abundant between the inflamed and noninflamed mucosa of CD patients. The histogram of the relative abundance of each group is shown on the left. In the middle is the log2 value of the average relative abundance ratio of the same genus in the two groups. The figure on the right shows the *P*-value and FDR values obtained by the Wilcoxon test. The dashed lines represent *P*-value = 0.05, or the value log2 (control/inflamed tissue) = 1.

## DISCUSSION

The involvement of the intestinal microbiome in IBD progression has been proven by previous studies (*Danne et al., 2024*). Large-scale data demonstrated differences in the intestinal flora composition of IBD patients in comparison with healthy controls (*Zheng et al., 2024*). There have been few studies on the bacterial microbiome profiles of inflamed mucosa in comparison with noninflamed mucosa in Crohn's disease patients.

In this study, we analyzed the composition and differences of intestinal bacteria from the inflamed mucosa of active CD patients in comparison with their noninflamed mucosa. Beta diversity and alpha diversity analyses demonstrated that there was no significant difference in the biodiversity of the bacterial microbiota in the inflamed mucosa compared with its noninflamed mucosa control. In a study of 14 patients with UC and 14 healthy controls, *Hirano et al. (2018)* discovered that microbial alpha diversity in both inflamed and noninflamed sites was significantly lower in UC patients than in non-IBD controls. Therefore, we inferred that it was plausible there was no significant difference in the biodiversity of bacterial microbiota between the inflamed and noninflamed sites of the CD patients analyzed in this study. If the diversity of inflamed and noninflamed sites of CD patients was lower than that of healthy controls, which is the case in UC patients, then their difference might not be great enough to be statistically significant. In the current study, the intestinal mucosa of CD patients was rich in *Bacteroidaceae* (*Bacteroides*), *Prevotellaceae* (*Prevotell*), *Fusobacteriaceae* (*Fusobacterium*), and *Enterobacteriaceae* (*Escherichia*). There was no significant difference in the community compositions at the family level between inflamed tissues and noninflamed tissues (Fig. 2). The Wilcox test results identified *Methylobacterium*, *Rothia*, *Capnocytophaga*, *Shinella*, *Actinomyces*, *Gardnerella*, *Leucobacter*, and *Bifidobacterium* as significantly different genera between inflamed and noninflamed mucosa of CD patients. The abundance of these genera was increased in the inflamed mucosa of CD patients relative to their noninflamed counterparts (Fig. 4D). Similar or contradictory reports exist for other differentially abundant genera that were found in the inflamed mucosa of CD patients from this study.

There were no reports associated with *Shinella* and *Leucobacter* on CD. In the high-dose Selena-l-methionine treatment group, *Shinella* increased in abundance. The expression of pro-inflammatory genes (including IL-1$\beta$ and IL-8) was significantly upregulated, while the expression of intestinal barrier-related genes (such as cdh1, ZO-1, ocln, and cldn7) was markedly suppressed (*Liu et al., 2021*). Therefore, *Shinella* may participate in the progression of CD by promoting the production of pro-inflammatory factors and increasing intestinal mucosal permeability. *Leucobacter* is a bacterial genus belonging to

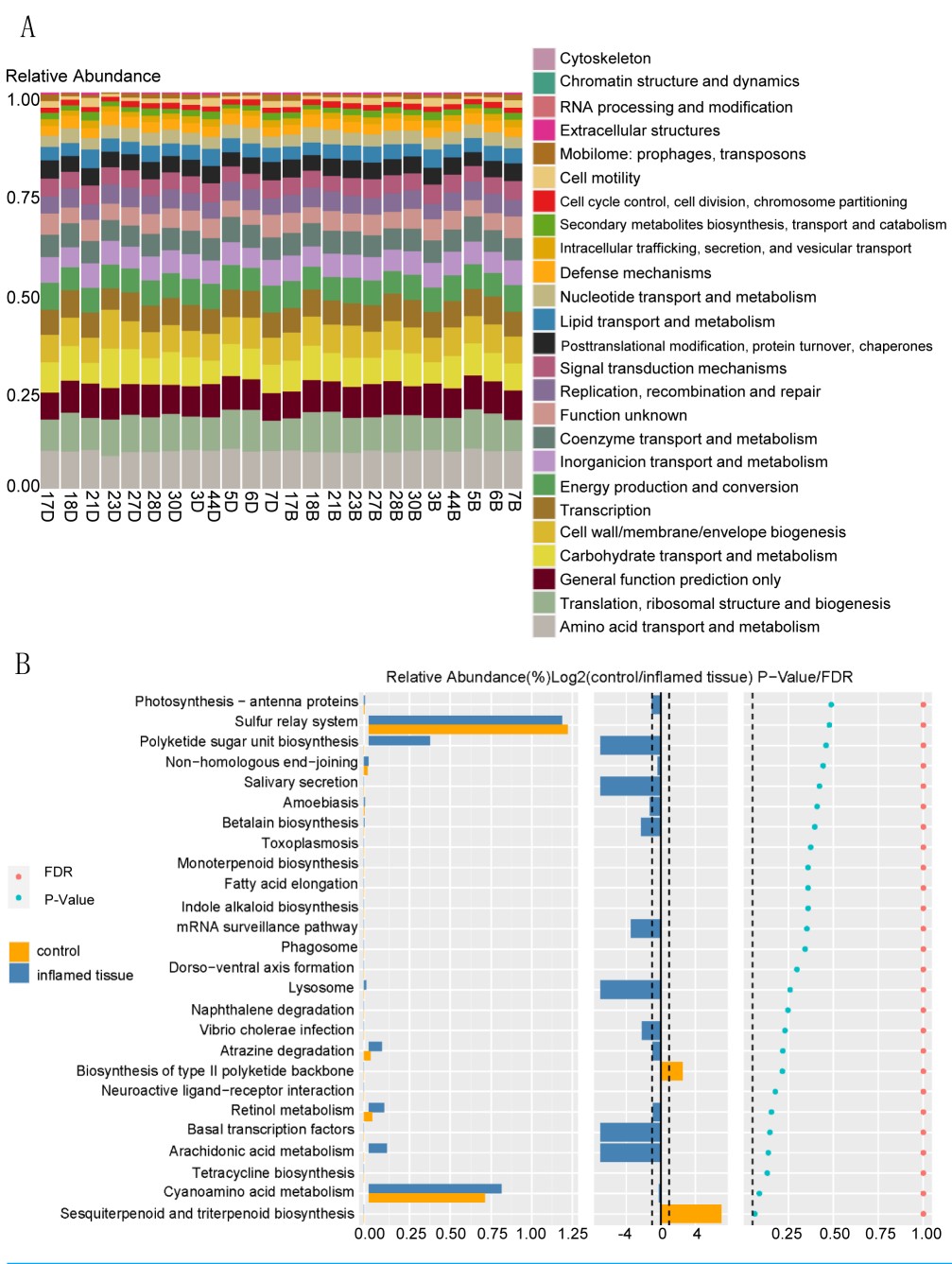

**Figure 5  Functional prediction of bacteria that are upregulated in abundance in inflamed mucosa.** (A) KEGG functional prediction shows the predicted function of the colony based on the KEGG database. The horizontal axis represents the sample, and the vertical axis represents the relative abundance of the predicted function. (B) Path difference Wilcox Test result graph. Functional prediction of bacteria with up-regulated abundance in inflamed mucosa indicated that there were no significantly different pathways between these two groups. The left side shows a histogram of relative abundance for each group. The middle displays the log2 value of the relative abundance mean ratio of the same channel in the two groups. The right side presents the $P$-value and FDR value obtained by the Wilcoxon test. The dashed lines represent $P$-value = 0.05, or the value log2 (control/inflamed tissue) = 1.

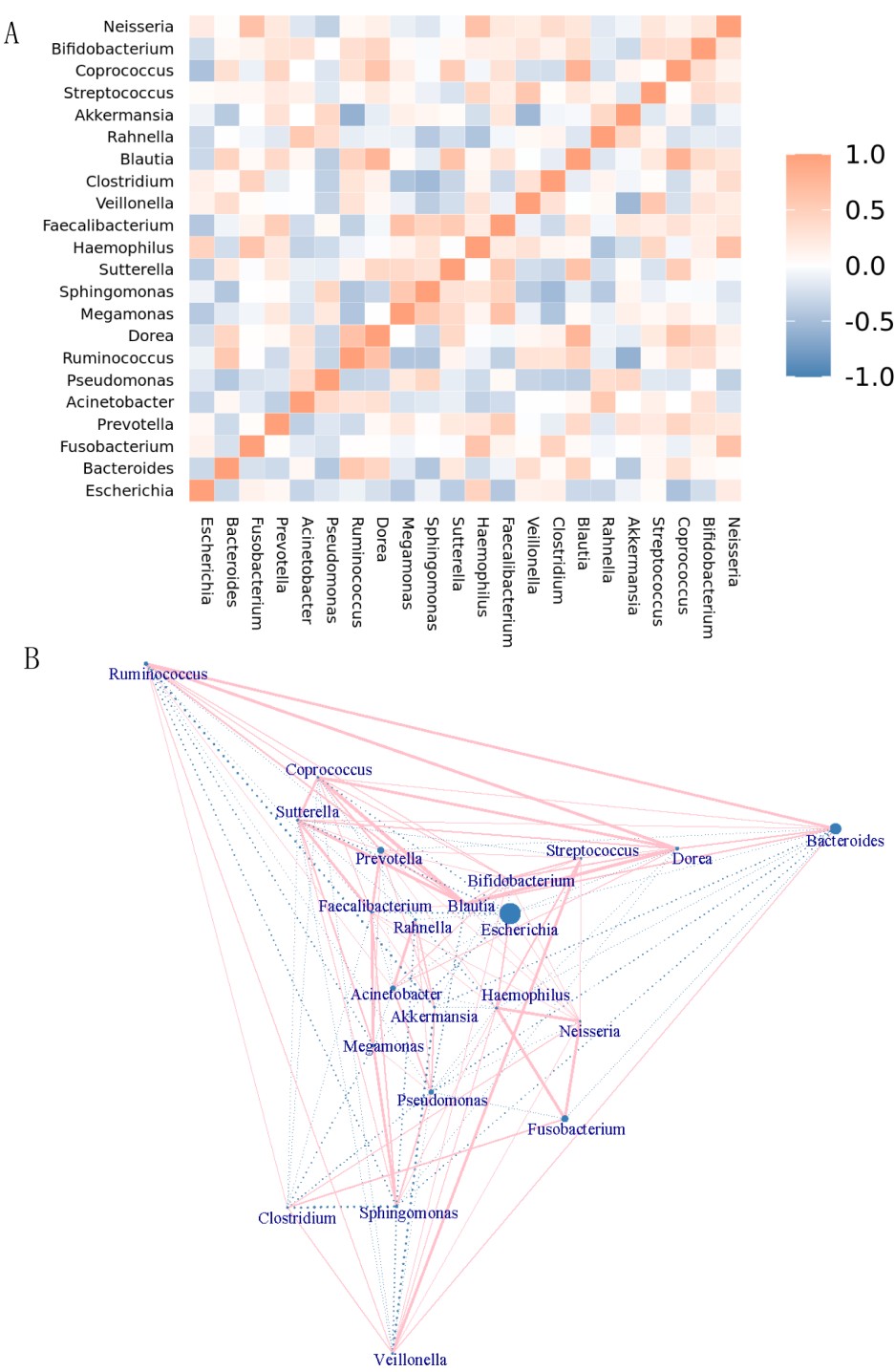

**Figure 6 Correlation analysis and model prediction.** (A) Spearman correlation coefficient analysis of the genus. The different genera were obtained by the rank sum test, and a Spearman correlation heatmap among dominant genera was drawn by R software. Through the heatmap, important patterns and relationships among dominant genera can be found. The graph shows correlations between different genera. A darker color indicates more closely correlated genera. (B) Genera network map. Each node in the graph (continued on next page...)

**Figure 6 (…continued)**
represents a genus. As the size of the circle increases, the average relative abundance of the genera also increases. Straight lines are used to link genus to genus, with pink indicating a positive correlation, blue indicating a negative correlation, and the thickness of the lines indicating the magnitude of the correlation. This figure only shows the relationship between genera with a correlation coefficient greater than 0.2.

the phylum *Actinobacteria*. There was no direct data on *Leucobacter*'s virulence factors or clinical cases in humans. Its Gram-positive structure and environmental prevalence suggest possible mechanisms akin to other opportunistic pathogens.

Increased abundance of *Actinomyces* was found in patients with immune-mediated inflammatory disease (including CD, UC, multiple sclerosis, and rheumatoid arthritis) and Crohn's-like appendicitis (*Forbes et al., 2018*; *Horvath et al., 2019*; *Nahum, Filice & Malhotra, 2017*; *Pittayanon et al., 2020*; *Ruigrok et al., 2021*; *Zhou et al., 2021*; *Zilberstein et al., 2025*). Generally, *Actinomyces* were considered commensal microorganisms in previous studies and caused opportunistic infections (*Turroni et al., 2008*). Therefore, we inferred that *Actinomyces* could invade the inflamed mucosa of patients with CD. Dysfunctional intestinal mucosal immunity in CD may enable *Actinomyces* to exhibit opportunistic pathogenicity. *Gardnerella* is considered to be a conditioned pathogen of vaginitis. A higher frequency of *Gardnerella* vaginalis biofilms was discovered in the urine of IBD patients (CD 38%, UC 43%) than in that of the control group (16%) (*Schilling, Loening-Baucke & Dörffel, 2014*). Compared with CD patients, patients with UC were significantly more enriched in *Gardnerella* in their fecal samples (*Zhou et al., 2021*). In this study, *Gardnerella* was discovered to be significantly more enriched in the inflamed mucosa of CD patients compared with their respective noninflamed mucosa. These data indicated that *Gardnerella* may participate in CD progression as a conditioned pathogen.

The relative abundance of *Methylobacterium* was increased in creeping fat taken from patients with Crohn's disease (CD) (*Sun et al., 2021a*). *Chiodini et al. (2015)* discovered that the abundance of *Methylobacterium* was specifically increased 6.1-fold in the submucosa compared with the mucosa in CD patients. One study by *Wang et al. (2021)* on pediatric CD patients discovered that a sustained response to IFX therapy was associated with a higher abundance of *Methylobacterium*. *Methylobacterium* may play important roles in triggering immune responses. *Methylobacterium spp.* are fastidious microorganisms that grow slowly and are tolerant to high temperatures, drying, cleaning, and disinfecting agents. Therefore, they can form biofilms and are predominant in hospital environments, particularly in endoscope channels and tap water (*Kovaleva, Degener & Van der Mei, 2014*). In our study, the abundance of *Methylobacterium* was enriched in the inflamed mucosa of CD patients compared with their respective noninflamed mucosa. CD patients frequently receive endoscopic inspection, which may lead to opportunistic colonization of *Methylobacterium*. CD is a disease characterized by persistent and overactive immune responses. Thus, the function of *Methylobacterium* in immune regulation should be further studied.

The genera *Capnocytophaga* and *Rothia* are opportunistic pathogens isolated frequently from oral cavities (*Fatahi-Bafghi, 2021*; *Idate et al., 2020*). There are few reports on the mechanisms by which *Capnocytophaga* may participate in immune regulation.

*Sun et al. (2021b)* reported that patients with both CD and periodontitis had a relatively higher abundance of *Rothia*, while healthy individuals had a relatively lower abundance of *Capnocytophaga* (*Yang et al., 2021*). Our results indicated an increased abundance of *Rothia* and *Capnocytophaga* in the inflamed mucosa of CD patients compared with the same patients' noninflamed mucosa. *Rothia* plays an important role in immunomodulation. Macrophages (*Bednár & Mára, 1991*) or lymphocytes (*Fotos, Gerencser & Gerencser, 1982*) can be activated by *Rothia* or antigens from *Rothia* in periodontal disease. Specific strains of *Rothia* such as *R. dentocariosa* ATCC14189 and ATCC14190 can induce TNF-alpha production in macrophages (*Kataoka et al., 2014*). There was persistent activation of lymphocytes and higher expression levels of TNF-alpha in CD patients. Accordingly, we inferred a similar immunomodulatory function of *Rothia* in CD.

Reports on the abundance of *Bifidobacterium* in IBD patients vary. Generally, the abundance of *Bifidobacterium* in patients with intestinal diseases is much lower than in healthy controls (*Arboleya et al., 2016*). The load of *Bifidobacterium* in fecal samples (ileum and colon) of patients with CD was decreased compared to that in healthy people (*Kowalska-Duplaga et al., 2019*; *Qiu et al., 2020*). *Forbes et al. (2018)* reported that the abundance of *Bifidobacterium* was higher in stool samples of patients with UC. An increased proportion of *Bifidobacterium* was discovered to be higher in fecal and biopsy samples of active CD patients or UC patients (*Wang et al., 2014*). In this study, there were no flora distribution data of mucosa from healthy people. However, the abundance of *Bifidobacterium* in the inflamed mucosa of CD patients was discovered to be relatively increased compared with that in their noninflamed mucosa. Few studies have revealed the load of *Bifidobacterium* in patients with CD compared with their noninflamed mucosa.

In a randomized trial, a diet low in fermentable oligosaccharides, disaccharides, monosaccharides, and polyols (FODMAPs) reduced specific symptoms, thereby reporting adequate symptom relief in quiescent IBD. Patients on the low FODMAP diet had a significantly lower abundance of *Bifidobacterium adolescentis* and *Bifidobacterium longum* in stool samples (*Cox et al., 2020*). Numerous studies have indicated that *Bifidobacterium* are capable of protecting against intestinal inflammation by enhancing the intestinal epithelial tight junction barrier and reducing inflammatory cytokine expression (*Yao et al., 2021*; *Al-Sadi et al., 2021*). *Bifidobacterium* strains were discovered to induce IL-10, IL-6, and MCP-1 expression in peripheral blood mononuclear cells (*Dong, Rowland & Yaqoob, 2012*). Therefore, we inferred a complex function of *Bifidobacterium* in CD progression. In this study, the load of *Bifidobacterium* in the inflamed mucosa of patients with CD was higher than that in their noninflamed mucosa. *Bifidobacterium* in inflamed mucosa may play a protective role by enhancing epithelial the tight junction barrier and inducing IL-10 expression. On the other hand, *Bifidobacterium* may promote inflammation by inducing proinflammatory cytokine secretion (IL-6, MCP-1, *etc*).

There were limitations in this study. Firstly, the sample size is small. Only 12 paired inflamed and noninflamed mucosa were included. Secondly, QIIME 1 (v1.8) was used in 16S rRNA analysis by OTU clustering rather than ASV approaches that provide single-nucleotide resolution. The taxonomic annotation reliability of 16S rRNA gene sequencing is limited. Species cannot been detected accurately using short-read sequencing. Therefore,

the authors made the analysis stop at the genus level. Thirdly, the analysis failed to address potential contamination issues inherent in tissue microbiome studies, which is critical for data interpretation. Negative controls should be incorporated in subsequent studies.

## CONCLUSIONS

In this study, we found the following genera *Methylobacterium, Rothia, Shinella, Capnocytophaga, Actinomyces, Gardnerella, Leucobacter,* and *Bifidobacterium* to be significantly enriched in the inflamed mucosa of active CD patients compared with their noninflamed mucosa. Opportunistic pathogens may play important roles in the immunoregulation of CD progression. The findings from this study provide new evidence for the viewpoint that the dysbiosis of mucosa-associated microbiota contributes to CD development, from a self-comparison perspective. While the cohort size is limited, the findings nonetheless provide an important foundation for future, larger-scale investigations.

### Funding
This work was supported by the Science and Technology Plan of Suzhou Citizens' Health (No. SKJY2021123), the Suzhou Special Project of Diagnosis and Treatment for Key Clinical Disease (No. LCZX201715), and the Natural Science Foundation of Jiangsu Province (No. BK20161232), The Gut Microbiota Research Project of Gusu School, Nanjing Medical University (No. GSKY20240804). The funders had no role in study design, data collection and analysis, decision to publish, or preparation of the manuscript.

### Grant Disclosures
The following grant information was disclosed by the authors:
Science and Technology Plan of Suzhou Citizens' Health: SKJY2021123.
Suzhou Special Project of Diagnosis and Treatment for Key Clinical Disease: LCZX201715.
Natural Science Foundation of Jiangsu Province: BK20161232.
The Gut Microbiota Research Project of Gusu School, Nanjing Medical University: GSKY20240804.

### Competing Interests
The authors declare there are no competing interests.

### Author Contributions
- Juan Yin conceived and designed the experiments, performed the experiments, analyzed the data, prepared figures and/or tables, authored or reviewed drafts of the article, and approved the final draft.
- Tong Hu performed the experiments, analyzed the data, prepared figures and/or tables, and approved the final draft.

- Liping Zhang analyzed the data, prepared figures and/or tables, and approved the final draft.
- Lijuan Xu analyzed the data, prepared figures and/or tables, and approved the final draft.
- Jianyun Zhu analyzed the data, authored or reviewed drafts of the article, and approved the final draft.
- Yulan Ye conceived and designed the experiments, authored or reviewed drafts of the article, and approved the final draft.
- Zhi Pang conceived and designed the experiments, authored or reviewed drafts of the article, and approved the final draft.

### Human Ethics

The following information was supplied relating to ethical approvals (i.e., approving body and any reference numbers):

The Ethics Committee of Nanjing Medical University approved the study (K2021-064-K01).

### Data Availability

Raw data is available in the Supplemental Files.

Data is available at GenBank: PRJNA923103.

### Supplemental Information

Supplemental information for this article can be found online at http://dx.doi.org/10.7717/peerj.19959#supplemental-information.

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
