# Peer review of "Bacterial microbiome profiles of the inflamed terminal ileum mucosa in active Crohn's disease patients"

_PeerJ, doi:10.7717/peerj.19959_

## Round 0.1 · original submission · Major Revisions

All three reviewers had major concerns about the manuscript, particularly around the sample sizes, analysis pipelines and the data not having been submitted to a public repository.

**Language Note:** The review process has identified that the English language must be improved. PeerJ can provide language editing services - please contact us at [email protected] for pricing (be sure to provide your manuscript number and title). Alternatively, you should make your own arrangements to improve the language quality and provide details in your response letter. – PeerJ Staff

Reviewer 1 ·

Basic reporting

The language appears generally clear and professional, though a thorough review for any grammatical or phrasing issues is recommended for clarity. The introduction provides good background for the study, highlighting the relevance of the gut microbiome in CD and the limitations of previous studies. It can be improved by adding more information about CD, parts of the gut involved
Relevant literature is referenced throughout the paper. Figures and tables are included to present the results, and their quality and labeling should be checked according to the guidelines. Authors must make sure to have the names of the bacteria italicized (e.g. in the abstract)

Experimental design

1. The research question is clearly defined: to characterize the bacterial microbiome profiles of inflamed mucosa in comparison with noninflamed mucosa from Crohn's disease patients.
2. The study design involves collecting paired inflamed and noninflamed mucosal biopsies from the same patients, which is a strength for self-comparison.
3. Sample size (n=12), is low, although it may be hard to find a lot of patients with CD, who agree to provide mucosal tissues.
4. Authors must provide information about hospitals, from where patients were recruited in the methods. Alos, it is important to mention the duration of sample collection.
5. Methods for sample collection, DNA extraction, 16S rRNA gene sequencing, and data analysis (OTU clustering, species annotation, diversity analysis, differential composition analysis, functional prediction, and correlation analysis) are described. The level of detail should be assessed for replicability.
6. Ethical approval and informed consent are mentioned. The ethical conduct of the experiments should be explained in detail. Were the samples collected for research purposes only, or as part of the routine diagnostic workflow?

Validity of the findings

1. Key findings include the identification of significantly enriched genera in inflamed mucosa and the observation that overall alpha and beta diversity were not significantly different between inflamed and noninflamed sites within CD patients.
2. Statistical methods like Wilcoxon test and LEfSe are mentioned for analyzing differences and identifying biomarkers. The soundness of the statistical analysis should be evaluated.
3. The discussion interprets the findings in the context of previous research, noting both consistencies and contradictions. Potential roles of the differentially abundant bacteria must be further discussed.
4. The conclusion summarizes the key findings and suggests that opportunistic pathogens may play roles in immunoregulation in CD. The authors also appropriately note a limitation of the study (sample size) and suggest future research with a larger sample size.

Additional comments

The paper presents a focused study with a clear objective and relevant methodology. The self-comparison approach using paired samples from CD patients is a strength. The findings identify specific bacterial genera associated with inflamed mucosa, contributing to the understanding of microbial dysbiosis in CD.
Points for Improvement/Further Consideration
• A thorough review of the English language for clarity and precision is recommended.
Notes:
Line 58: gut mycobiome is mostly “microbiome”
Line 77-79: Even if the mucosa in CD is not inflamed, can you role out the possibility of the disease in this part of the gut, as CD affects any part of the gut form mouth to anus.
Line 82: Recruitment of participants and collection of terminal ileum mucosal samples: this section must be divided into 2 parts, one for recruitment and another for sample processing. Also, the site of sample collection is both trenail illum and colon, thus both must be mentioned in the section heading “terminal ileum mucosal samples” was mentioned without including colon.
Line 122: Species annotation, must be changed, as species can not detected accurately using short read sequencing; and analysis must stop at the genus level.
Line 170: age 27 to 53 years old, is this usual for treatment naive CD? It usually affects younger people; can this be explained?
Figure 3 C, correct “imflammed” into inflamed , in figure label , also in Fig 4,please check all for spelling error.

Reviewer 2 ·

Basic reporting

This study investigates microbial differences between inflamed and non-inflamed ileal mucosa in CD patients, which represents an interesting scientific question. However, the background section fails to adequately contextualize the study within existing knowledge. Key prior studies (e.g., PMID: 17262808, 32448900, 37172588) demonstrating microbial-host interactions in ileal Crohn’s disease were omitted. This oversight undermines the manuscript’s ability to articulate its novelty and incremental contributions to the field. The manuscript does not comply with open science standards. Raw sequencing data must be deposited in a public repository (e.g., SRA accession SRPXXXXXX, NODE project OEPXXXXXX) prior to publication, with accession numbers included in the methods section. Some figures with poor resoultion

Experimental design

1、Insufficient sample size: The study includes only 12 paired samples, which is inadequate to draw statistically robust conclusions. Previous studies (e.g., PMID: 17262808, 32448900,37172588) have already examined host-microbiome interactions in inflamed vs. non-inflamed tissues. The novelty and incremental contribution of this work remain unclear.

Validity of the findings

2. Methodological limitations: The 16S rRNA analysis still employs outdated OTU clustering rather than current ASV approaches that provide single-nucleotide resolution. Functional predictions rely on PICRUSt1 instead of the updated PICRUSt2 pipeline. The analysis fails to address potential contamination issues inherent in tissue microbiome studies, which is critical for data interpretation.
3. The Results section frequently describes figures rather than synthesizing key observations (e.g., Lines 201-209). This narrative approach obscures the core findings and their biological implications.

Additional comments

4.minor issues:
Line 58: "mycobiome" should likely be "microbiome" unless fungi were specifically analyzed.
Line 188: Typo "bokth" → "both".
Line 189: Clarification needed for parenthetical values (e.g., Escherichia coli (0.2771, 0.2903)) – are these relative abundances? Coordinates?
Line 193: The term "bacterial typing" is ambiguous – specify whether this refers to strain typing, taxonomic classification, or another method.
5、It is recommended that the raw sequencing data be deposited in a publicly accessible repository such as the Sequence Read Archive (SRA) or the National Omics Data Encyclopedia (NODE) to ensure data transparency, reproducibility, and compliance with journal requirements.

Reviewer 3 ·

Basic reporting

This study presents a well-designed investigation into the spatial distribution of the gut microbiome in the context of mucosal inflammation in Crohn’s disease—a key area that remains relatively underexplored. The authors clearly articulate their aims, which addresses a significant knowledge gap in the field. A notable strength of the study is the inclusion of treatment-naïve Crohn’s disease patients with active disease, which adds valuable insight into the early-stage microbial and inflammatory landscape. While the cohort size is limited, the findings nonetheless provide an important foundation for future, larger-scale investigations, which should be reiterated in the conclusions.

The discussion is generally well-written and provides a thoughtful interpretation of the study’s findings, particularly in relation to potential biomarker taxa. The authors do a commendable job of integrating current literature to support their conclusions.

However, the manuscript would benefit from more thorough proofreading. For instance, figure legends (e.g., Figure 3C) and bacterial names contain spelling errors.

Experimental design

The study employs the RDP Classifier v2.2 and QIIME v1 for microbial taxonomic classification, which, while widely used historically, are now considered outdated. I strongly recommend updating the analysis pipeline to include more current tools such as QIIME2, the SILVA database, and PICRUSt2. These tools offer improved taxonomic resolution and more accurate functional predictions, which could strengthen the study’s conclusions.

Notably, recent updates to reference databases have reclassified members of Faecalibacterium prausnitzii into distinct species. This raises the possibility that re-analysis with newer databases could lead to shifts in the reported taxonomic composition and potentially refine or alter the study’s findings.

The packages used for alpha diversity calculation method is not clearly specified. Please indicate the software or statistical package used for alpha diversity analysis.

Validity of the findings

It would also be valuable to report the average relative abundance of Faecalibacterium within this cohort. Given that all subjects were treatment-naïve patients with active Crohn’s disease, one would expect a marked reduction compared to healthy controls. Providing this comparison would add important biological context to the microbiome findings.

In Figure 4A (LEfSe analysis), the presentation of results could be clarified. It is unclear whether entire orders (e.g., Bifidobacteriales) or specific unclassified members within these taxa were found to be associated with inflamed mucosa. Please specify the taxonomic resolution of these associations to avoid misinterpretation.

I am curious as to whether the microbiome profiles demonstrated any patient-specific clustering. It is commonly observed that inter-patient variability exceeds intra-patient variability in microbiome studies.
Inclusion of an unsupervised PCoA plot of the microbial profiles (e.g., based on UniFrac distances) would help visualize the overall distribution and variation among samples, and strengthen the ecological interpretation of the data.

---

## Round 0.2 · accepted · Accept

The reviewer was happy that the authors had made a genuine attempt to address all previous concerns. The submission of raw data is also welcomed.

Reviewer 3 ·

Basic reporting

In response to reviewer comments highlighting the need for improved language throughout the manuscript, I commend the authors for their efforts in revising the text to address spelling and grammatical issues. These revisions have significantly enhanced the clarity and readability of the manuscript compared to the initial submission.

Experimental design

It is regrettable that the co-authors were unable to utilize more recently developed 16S rRNA gene profiling tools. Nonetheless, I appreciate their efforts to engage with their sequencing provider in an attempt to reanalyze the data using updated pipelines. As additional samples are collected, I encourage the authors to consider re-sequencing both the current and future samples, and to analyze them using at least the QIIME2 bioinformatics pipeline in conjunction with updated 16S rRNA gene reference databases such as SILVA.

Validity of the findings

Overall, this study remains one of the few to investigate the mucosa-associated microbiome in a Crohn’s disease cohort, while also performing a pairwise comparison of microbial profiles between inflamed and non-inflamed tissue within the same individuals. The authors have also more clearly acknowledged the study’s limitations and addressed several of the concerns I previously raised, which, in my view, has strengthened and better substantiated the manuscript.